# Seroprevalence of SARS-CoV-2 Antibodies and Associated Factors in Healthcare Workers before the Era of Vaccination at a Tertiary Care Hospital in Turkey

**DOI:** 10.3390/vaccines10020258

**Published:** 2022-02-08

**Authors:** Meliha Cagla Sonmezer, Enes Erul, Taha Koray Sahin, Ipek Rudvan Al, Yasemin Cosgun, Gulay Korukluoglu, Humeyra Zengin, Gülçin Telli Dizman, Ahmet Cagkan Inkaya, Serhat Unal

**Affiliations:** 1Infectious Diseases and Clinical Microbiology Department, Hacettepe University Faculty of Medicine, 06100 Ankara, Turkey; ipekridvanal@hacettepe.edu.tr (I.R.A.); humeyra@hacettepe.edu.tr (H.Z.); gulcin.telli@hacettepe.edu.tr (G.T.D.); inkaya@hacettepe.edu.tr (A.C.I.); sunal@hacettepe.edu.tr (S.U.); 2Internal Medicine Department, Hacettepe University Faculty of Medicine, 06100 Ankara, Turkey; eneserul@hacettepe.edu.tr (E.E.); koraysahin@hacettepe.edu.tr (T.K.S.); 3National Arboviruses and Viral Zoonotic Diseases Laboratory, Microbiology Reference Laboratories Department, Public Health General Directorate of Turkey, 06100 Ankara, Turkey; yasemin.cosgun@saglik.gov.tr (Y.C.); fatma.korukluoglu@saglik.gov.tr (G.K.)

**Keywords:** COVID-19, healthcare workers, before vaccination, seroprevalence, SARS-Co-V 2 antibody

## Abstract

Healthcare workers (HCWs), as frontliners, are assumed to be among the highest risk groups for COVID-19 infection, especially HCWs directly involved in patient care. However, the data on the COVID-19 infection and seroprevalence rates are limited in HCWs. Therefore, we aimed to evaluate the seroprevalence rates in HCWs according to risk groups for COVID-19 contraction in a large cross-sectional study from a tertiary care hospital in Turkey. We enrolled 1974 HCWs before the vaccination programs. In two separate semi-quantitative ELISAs, either IgA or IgG antibodies against SARS-CoV-2 spike protein subunit 1 (S1) were measured. The proportion of positive test results for IgG, IgA, or both against SARS-CoV-2 of study subjects was 19% (375/1974). Frontline HCWs who had contact with patients (21.7%, RR 2.1 [1.51–2.92]) and HCWs in working in the COVID-19 units, intensive care units, or emergency department (19.7%, RR 1.61 [1.12–2.32]) had a notably higher Anti-SARS-CoV-2 IgG compared to the rest of HCWs who has no daily patient contacts ([11.1%]; *p* < 0.0001). HCWs who care for regular patients in the medium-risk group have also experienced a sustained higher risk of exposure to SARS-CoV-2. We should enhance the precaution against COVID-19 to protect HCW’s safety through challenging times.

## 1. Introduction

Towards December 2019, a novel coronavirus was first identified in a cluster of pneumonia cases in Wuhan, Hubei Province, China. On 11 March 2020, the World Health Organization (WHO) had declared the novel coronavirus outbreak as a global pandemic [1,2]. As of 7 September 2021, more than 200 million cases of COVID-19 have been documented globally, and the pandemic has left close to 4.6 million people dead [3,4]. While coronavirus disease 2019 (COVID-19) remains a serious concern, front-line healthcare workers (HCWs) are one of the highest-risk occupational groups for COVID-19 infection, since they have contact with both COVID-19 patients and other healthcare professionals [5,6]. Initial estimates suggest that front-line HCWs may account for 10–20% of all COVID-19 diagnoses, and subsequent calculations based on meta-analyses of antibody prevalence from several countries demonstrate that the presence of IgG and/or IgM antibodies among HCWs has been found to vary between 7% and 9% [7,8,9,10]. In addition, it has been suggested that HCW may have a higher viral load and worse clinical outcomes than the general population due to repeated exposure to the virus. Still, the available data are conflicting [5,6,11].

It should be noted, however, that the risk of infection is not the same in every HCW group. Some HCW groups have a higher risk of infection than others with a medium risk of infection [5]. Apart from patient contact, risk factors associated with seroprevalence are still uncertain. Furthermore, it remains unclear whether individual or work-related characteristics, including occupation with direct patient care or regular patient contact, increase the risk of COVID-19 infection [8,9,12,13].

Seroprevalence of SARS-CoV-2 antibodies among HCWs must be determined to identify the level of exposure among HCWs, know high-risk groups in HCWs, and explain the transmission of COVID-19 among HCWs. It might also help understand the asymptomatic spread of SARS-CoV-2 and the infection control committee’s efficacy in preventive measures [1,2,7,14,15]. This paper aimed to investigate the seroprevalence of SARS-CoV-2 antibodies in a random sample of HCWs from a large tertiary care hospital and determine risk factors associated with it among HCWs. The secondary aim is to estimate the baseline prevalence of SARS-CoV-2 carriage among previously undiagnosed and asymptomatic HCWs.

## 2. Materials and Methods

### 2.1. Study Design

In this study, we invited HCWs for serological testing within the scope of surveillance who worked at Hacettepe University Hospitals (including physicians, nurses, lab technicians, medical students, etc.). We recorded participant characteristics, history of medical conditions, demographics, occupational factors, family history of COVID-19 disease, symptoms that might be related to infection, and results of PCR testing. Additionally, we asked participants to complete a questionnaire and donate a venous blood sample for serologic testing from 15 October 2020, to 18 October 2020. The study was performed at an early pandemic stage before the vaccination program in Turkey when serum antibodies for SARS-CoV-2 could not be routinely performed outside the research. In total, 1974 HCWs were recruited for this study.

HCW who has seroposivity against SARS-CoV-2 with no previous history of COVID-19 diagnosis and/or symptoms related to COVID-19 infection noted to have asymptomatic infection.

### 2.2. Samples Collection and SARS-CoV-2 Antibody Testing

The presence of IgA and IgG antibodies against SARS-CoV-2 was tested concurrently in the blood samples collected from the HCWs. The blood samples taken into sterile gel blood tubes were centrifuged at 3500 RPM for 10 min to obtain serum samples. Serum samples were heat-inactivated at 56 °C for 30 min, the rest serum samples were transferred to 2 mL cryovial microtubes, and the samples stored at 4 °C were used within five days. Before the ELISA testing, all models were held at room temperature and centrifuged briefly. The Anti-SARS-CoV-2 IgG and IgA ELISAs (Euroimmun Medizinische Labordiagnostika, Lübeck, Germany; Cat # EI 2606-9601 G and EI 2606-9601 A) were performed according to the manufacturer’s instructions. In two separate semi-quantitative ELISAs, IgA or IgG antibodies against SARS-CoV-2 spike protein subunit 1 (S1) are detected in human serum. According to the manufacturer’s recommendations, the IgG and Ig A index of serum (100 µL) samples with a value of 0.8 are negative, 0.8 and 1.1 borderline, and 1.1 positives. However, 1.1 was utilized as a more strict cut-off value for positive results for sensitivity and specificity, and all values <1.1 were considered negative [16].

### 2.3. Risk Categories of Participants

Participants were separated into three risk categories (high, medium, and low) according to their work activity.

High-risk: Units in daily contact with COVID-19 patients, working with aerosol-generating procedures (pandemic service, pandemic intensive care, pandemic outpatient clinic, and emergency department).

Medium-risk: Internal and surgical services/outpatient clinics, radiology, laboratory, security units in contact with non-COVID-19 patients.

Low-risk: Administrative and support areas with no daily patient contact.

Additionally, HCWs were separated according to the type of hospital units as intensive care units (ICUs), COVID units (units that cared for COVID-19 patients at some point throughout the study period except ICU), emergency department, and “other” units (including basic science, administrative and management units that did not have regular contact with patients).

### 2.4. Ethical Approval

The protocol was conducted in agreement with the Helsinki Declaration. Written informed consent was obtained from each participant. The ethics committee approval was obtained from Hacettepe University Observational Research Ethics Committee (Ethics Committee Approval No: 2021/13-37).

### 2.5. Statistical Analysis

Statistical analyses were performed using the SPSS software version 22. Categorical variables were expressed as counts and percentages. Continuous variables such as age were expressed as mean and standard deviation (SD). Seroprevalence of SARS-CoV-2 antibodies was reported as a percentage, and the 95% confidence interval (CI) was estimated using a finite population correction. The Chi-square test or Fisher’s exact test (when chi-square test assumptions do not hold due to low expected cell counts), where appropriate, was used to compare antibody test results in different groups. Multivariable logistic regression that included statistically significant risk factors in the univariable analysis was performed, examining the association between risk groups and serology positivity and adjusted for the age and sex. A *p*-value of less than 0.05 was considered to show a statistically significant result.

## 3. Results

The enrolled HCWs were 37.8 ± 9.3 years on average, ranging from 18 to 65, and 64.9% (1282/1974) were women. While the SARS-CoV-2 Ig G seropositivity rate was similar (15.6–19.7%) between in the <29, 30-39, 40–49 and 50–59 age ranges (*p* = 0.547), numerically the 60 < age HCW group had the lowest seropositivity (12%). The SARS-CoV-2 Ig G seropositivity rate was similar in female (18.2%) and male (19.1%) HCWs (*p* = 0.623). Comorbidities were fairly low in our study population; 3% (69/1974) reported asthma, Chronic obstructive pulmonary disease (COPD), Type 2 Diabetes, or atherosclerotic heart disease.

In terms of participants’ roles in the healthcare setting, the most common professions were nurses (25.4%, 502/1974), followed by medical doctors (16.6%, 328/1975), and healthcare assistants (15.9%, 313/1974). The highest SARS-CoV-2 Ig G seropositivity rate was observed in employees working in the kitchen service (26%) and healthcare assistants (19.5%). Furthermore, the lowest seropositivity rates were found in care workers (13%) and medical students (14.9%).

Regarding the type of hospital unit, the vast majority (7.3%) of HCWs had participated in the study from the emergency unit. The other units were as follows, in order of frequency; 4.3% (85/1974) ICU, 1.5% (30/1974) COVID-19 units, and 86.8% (1714/1974) other departments of the hospital. Baseline demographic characteristics of HCWs are shown in Table 1.

The highest seropositivity was found at 26.7% in HCW working COVID-19 units. Seropositivity of HCWs working in the emergency department and other departments (18.6–18.7%) was found similarly. Among HCWs who work in ICU (11.8%), COVID-19 exposures were lower unexpectedly.

There were no significant differences in SARS-CoV-2 Ig G antibodies seropositivity according to age, gender, the professional role of HCWs and, the type of hospital unit.

The proportion of positive test results for IgG, IgA, or both against SARS-CoV-2 of study subjects was 19% (375/1974). Of these, 4% (80/1974) had developed IgA, and 18.5% had developed IgG antibodies. In addition, 3.5% (70/1974) healthcare workers had developed both IgG and IgA antibodies.

This study showed that the prevalence of SARS-CoV-2 IgG seropositivity among asymptomatic HCWs was 8.7% (171/1974). Although 1.9% (38/1974) HCWs had negative for SARS-CoV-2 IgG diagnosed with COVID-19 disease in the medical history. 2.4% HCW (47/1974) became infected after the study period; almost 1/3 of them (15/1974) had the previous IgG against SARS-CoV-2.

Univariate and multivariate analyses for risk factors for SARS-CoV-2 seropositivity of HCWs are shown in Table 2. The seroprevalence of healthcare workers with a family history of COVID-19 disease was significantly higher (41.4%, RR 4.64 [3.57–6]) compared to those without a family history of COVID-19 (13.3%, *p* < 0.0001). HCWs working on the low-risk group who had no patient contacts daily had significantly lower seropositivity (11.1%) than other frontline HCWs (21.7%, RR 2.1 [1.51–2.92]) and contact with many COVID-19 patients (19.7%, RR 1.61 [1.12–2.32]; *p* < 0.0001).

## 4. Discussion

HCWs with direct patient contact are presumed to be at higher risk for SARS-CoV-2 infection. In contrast, the SARS-CoV-2 contraction in HCWs with lower occupational exposure risks is still vital to prevent any disruptions in inpatient care. Seroprevalence studies aimed to understand herd immunity, asymptomatic cases, and risk factors associated with infection. At the timeframe of our study, the vaccination program in HCWs had not yet started. Towards the end of 2020, vaccination programs began, and protection against disease was demonstrated, but the emergence of new variants is still a concern [17,18]. While primary prevention remains a central strategy against the pandemic, infection control measures such as compliance to the N 95, face shield, gown, gloves, and social distance over time might degrade. Therefore, studies on seroprevalence among HCWs maintain their importance to determine risk factors and take preventive measures.

Serological tests are generally used to diagnose patients who have negative PCR and computed tomography (CT) despite clinical suspicion, and to detect previous or ongoing infection. COVID-19 patients have been shown to develop IgM, IgA, and IgG against the virus’s S (spike) glycoprotein and N (nucleocapsid) proteins within two weeks of the onset of symptoms [19,20,21].

Early data showed that based on a lot of modeling, roughly half (44%) of the transmission probably occurs from asymptomatic and presymptomatic transmission [22]. In also our study, there were a substantial percentage of individuals with asymptomatic infection. Notably, this finding also emphasizes the importance of HCWs’ adherence to the precautionary preventive measures consisting of mask use, maintaining social distance and hand hygiene, regardless of the presence of signs of respiratory system infection, to prevent in-hospital transmission.

We have found that seropositivity in HCWs was highest in COVID 19 units (26.7%) and the lowest in critical care units (11.8%, *p* = 0.141). Although this difference was not statistically significant, it supports the results of other studies in the literature [23,24,25]. There may be many reasons for low antibody levels in HCWs in intensive care. To name a few: healthcare professionals in the ICU might be more trained and prepared for the pandemic, and adherence to infection preventive measures might be higher than other HCWs [26]. Additionally, patients in the ICU are more likely to have reduced symptomatic viral spread due to the late phase of disease progression [27].

HCWs with a family history of COVID-19 disease had higher seroprevalence (41.4%, RR 4.64 [3.57–6], *p* < 0.001) when compared to remaining HCWs. It is unclear whether the source of infection is HCWs or family members. It might be assumed that HCWs are more exposed to SARS-CoV-2. Then, therefore, it may make family members a vulnerable group to infection.

Notably, in our research, employees working in the kitchen service had the highest SARS-CoV-2 seropositivity rate (26%). In contrast, medical students had one of the lowest seropositivity among HCWs (14.9%); but it was statistically insignificant (*p* = 0.939). The reason beyond this difference may be that during our study’s timeframe, HCWs were allowed to eat their lunch and dinner in the hospital cafeteria. There might be a high risk for asymptomatic transmission for employees working in the kitchen service because HCWs do not wear masks in this mealtime indoor space. Medical students were permitted to take medical education from home online, and they were in the lowest risk group who had no daily patient contacts. As previously demonstrated, online education prevents COVID-19 transmission both in the community and in hospitals [28].

The seropositivity of HCWs has a significant variation between countries. While this rate was found to be 4% in China, where strict public health measures such as isolation and quarantine were applied along with adherence to infection control measures, it was found as 17.8% in a study in the USA similar to our research [6,29].

In a study involving 29,925 healthcare professionals in Denmark, similar to our research, the SARS-CoV-2 antibody levels in HCWs who contact patients (779 [4.55%] of 16,356) was significantly higher than other healthcare professionals who were not in contact with patients (384 [3.29%] of 11,657; RR 1.38 [1.22–1.56]; *p* < 0.001) [5]. Seroprevalence level was higher in male healthcare professionals than females; it was suggested that these differences might be due to female healthcare professionals being more likely to follow recommendations [5]. In our study, antibody levels were similar between genders (*p* = 0.623).

Our research is one of the first and most extensive studies to screen the seroprevalence of IgA and IgG antibodies against SARS-CoV-2 among HCWs in Turkey. Our main finding in the study is that HCWs in the medium (OR 2.1 (95% CI 1.51–2.92) and high-risk group (OR 1.61 (95% CI 1.12–2.32)) who made contact with patients had more than a two-fold higher risk of infection than HCWs in the low-risk group who had no daily patient contact. HCWs in the high-risk group, despite being in close contact with numerous COVID-19 patients and working with aerosol-generating procedures, have lower seroprevalence than HCWs in the medium-risk group. Data in the literature show that proper infection control practices, as well as the use of universal masking, can lower the seropositivity in HCW [30,31]. The reason for these results in our study as well may be differences in hygiene compliance. Further research is needed to evaluate the factors related to seropositivity in HCWs including questions about any unprotected exposures to a person with COVID-19.

There are several limitations to this study. The first is that it is based on data from a single center. The second limitation is the examination of antibody titers at a single time point and the inability to analyze variations in antibody levels over time. Another constraint is that many employees were assigned to different locations other than their units during the pandemic process and that comparisons between different branches of our hospital may be misleading. The fourth limitation of the study is that it is unknown whether healthcare professionals with a family history of COVID-19 disease are households with them, how often they meet, and whether they use personal protective equipment in contacts.

## 5. Conclusions

Healthcare workers in contact with the patient are at a higher risk for COVID 19 disease. Although the developments in antiviral treatment and vaccines, the primary and most crucial principle in protecting HCWs must be the widespread use of recommended personal protective equipment (such as masks, gloves, gowns, and eyewear). Adherence to personal protective equipment and environmental hygiene enhances the safety of HCWs.

## Figures and Tables

**Table 1 vaccines-10-00258-t001:** Baseline demographic characteristics of healthcare workers (HCWs).

	HCWs—*n* (%)	Seropositive HCWs—*n* (%)
**Age**		
≤29	474 (24)	88 (18.6)
30–39	690 (24.5)	137 (19.9)
40–49	561 (28.4)	102 (18.2)
50–59	224 (11.3)	35 (15.6)
≥60	25 (1.3)	3 (12)
**Sex**		
Female	1282 (64.9)	233 (18.2)
Male	692 (35.1)	132 (19.1)
**Profession**		
Physician	328 (16.6)	62 (18.9)
Nurse	502 (25.5)	94 (18.7)
Healthcare assistant	313 (15.9)	61 (19.5)
Technician	242 (12.3)	39 (16.1)
Administration staff	173 (8.8)	31(17.9)
Secretary	129 (6.5)	24 (18.6)
Medical students	74 (3.7)	11 (14.9)
Employees working in the kitchen service	50 (2.5)	13 (26)
Care workers	23 (1.2)	3 (13)
Other personnel	136 (6.9)	25 (18.3)
**Risk Categories**		
Low risk	503 (25.5)	56 (11.1)
Medium risk	929 (47.1)	202 (21.7)
High risk	542 (27.5)	107 (19.7)
**Type of Hospital Unit**		
Other	1714 (86.8)	320 (18.7)
COVID-19 units	30 (1.5)	8 (26.7)
Intensive care unit (ICU)	85 (4.3)	10 (11.8)
Emergency	145 (7.3)	27 (18.6)
**Family History of COVID-19 Disease**		
Yes	361 (18.3)	151 (41.8)

**Table 2 vaccines-10-00258-t002:** Univariate and multivariate analyses for risk factors for SARS-CoV-2 seropositivity of HCWs.

	Univariate Analysis			Multivariate Analysis		
Predictive Variable	OR	95% CI	*p*-Value	OR	95% CI	*p*-Value
**Age**						
≤29	1.00		0.594	1.00		0.547
30–39	1.67	0.49–5.71		1.08	0.77–1.53	
40–49	1.81	0.53–6.15		1.02	0.7–1.49	
50–59	1.63	0.47–5.54		0.84	0.51–1.37	
≥60	1.35	0.38–4.78		0.74	0.20–2.69	
**Sex**						
Female	1.00		0.623	1.00		0.623
Male	0.94	0.74–1.19		0.96	0.73–1.27	
**Profession**						
Physician	1.00		0.89	1.00		0.939
Nurse Nurse	0.98	0.69–1.41		0.92	0.62–1.37	
Healthcare assistant	1.03	0.7–1.53		1.04	0.67–1.61	
Technician	0.82	0.53–1.28		1.53	0.92–2.54	
Administration staff	0.93	0.58–1.5		1.6	0.93–2.8	
Secretary	0.98	0.68–1.65		1.87	1.01–3.46	
Medical students	0.74	0.37–1.5		0.88	0.41–1.9	
Employees working in the kitchen service	1.5	0.75-3		1.51	0.71–3.2	
Care workers	0.64	0.18–2.23		0.76	0.22–2.89	
Other personnel	0.96	0.57–1.61		1.64	0.92–2.93	
**Risk Categories**						
Low risk	1.00		<0.0001	1.00		<0.0001
Medium risk	2.21	1.61–3.05		2.1	1.51–2.92	
High risk	1.96	1.38–2.78		1.61	1.12–2.32	
**Type of Hospital Unit**						
Other	1.00		0.112	1.00		0.141
COVID-19 units	1.58	0.69–3.59		1.3	0.52–3.2	
Intensive care unit (ICU)	0.58	0.29–1.13		0.6	0.29–1,22	
Emergency	0.99	0.64–1.54		0.85	0.5–1.42	
**Family History of COVID-19 Disease**						
**Yes**	4.7	3.64–6.05	<0.0001	4.62	3.57–6	<0.0001

## Data Availability

The data that support the findings of this study are available on request from the corresponding author. The data is not publicly available due to privacy or ethical restrictions. Meliha Cagla Sonmezer; ORCID ID: 0000-0001-6529-5282.

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
