# Peer review of "Seroprevalence of SARS-CoV-2 Antibodies and Associated Factors in Healthcare Workers before the Era of Vaccination at a Tertiary Care Hospital in Turkey"

_vaccines, 2022, doi:10.3390/vaccines10020258_

Round 1

Reviewer 1 Report

The manuscript “Seroprevalence of SARS-CoV-2 Antibodies and Associated Factors in Healthcare Workers 2 Before the Era of Vaccination at a Tertiary Care Hospital” by Meliha Cagla-Sonmezer and colleagues provides and discuss findings accordingly to the aim and scope of the journal and in particular those related to seroprevalence rates in Healthcare workers (HCWs) according to risk groups for COVID-19 contraction in a large cross-sectional study from a tertiary care hospital.

The study investigates the seroprevalence of SARS-CoV-2 antibodies in a random sample of HCWs from a large tertiary care hospital and determine risk factors associated with it among HCWs and aims to estimate the baseline prevalence of SARS-CoV-2 carriage among previously undiagnosed and asymptomatic HCWs.

They performed serological tests by two separate semi-quantitative ELISAs in order to measure both IgA or IgG antibodies against the subunit 1 (S1) of the SARS-CoV-2 spike protein and they found that HCWs in the medium/high-risk had more than 2-fold higher risk of infection than HCWs in the low-risk group.

Although there are several typos and the narrative is kind of fragmentary, I just abide by scitnific soundness and here some points.

Line 78: not sure how the authors separated the serum, they should state that. Moreover, not sure whether the heat-inactivation interferes with Ab measurements (see doi: 10.1002/jcla.23411). I wonder if the heat inactivation might account for a potential underestimation of measuremnts hence affecting results and conclusions.

Line 129: not sure what the authors referrer to “the seropositivity rate”, they mean to IgA, IgG and/or IgM? In other words, how is calculated the “the seropositivity rate”?

Line 141-142: “Among HCWs who work in ICU (11.8%), COVID-19 exposures were lower unexpectedly” not sure, how tha authors demonstrate this?

Line 142: “This difference” which? It is not clear. Moreover, is redundant with line 144-145.

Did the authors calculated the gender difference within each age group and/or in regard to the risk level? Not sure whether the conclusions drawn about the gender are correct, they may change in regard to age class and/or risk level.

It is not clear whether or not the authors included those HCW with Family history of COVID-19 disease from the other experimental groups? This could be a bias and to “assumed that HCWs are more exposed to SARS-CoV-2 [lin 199]” I would suggest t repeat the analysis excluding those from the calculation.

Line 225: “…antibodies against SARS-CoV-2 among HCWs in Turke” this is the first time I read Turkey realizing only at the end of the MS that the study has been carried out in Turkey. This should be stated in the design of the study and may be useful to report in the abstract and title too.

Line 231: “We suggest that the difference in adherence to infection control measures among HCWs might be the most likely reason for this finding.” Not sure about this conclusion, how can the authors state that?

I would suggest to reanalyze some data in order to draw more solid conclusions and try to get a clearer and more convincing narrative.

Reviewer 2 Report

COPD, MI need to be spelled out

Type Diabetes is Type 2 Diabetes?

In related with definition of seropositive, authors may state vaccine not available in method, in addition to discussion.
Authors may breakdown type of hospital units with only those who actually work for care patients, likewise nurses or doctors. Otherwise it is quite hard to understand seropositive prevalence at other unit because the extent of exposure thru patient care is remarkably different. 

How do authors define asymptomatic? There is no description in method. In addition family history of COVID-19. Do author test families to know they had COVID-19 during study period? or self-answered history?

% of reported symptom among seropotive cases should be in result, not in dicussion.

High seroprevalnce in the kitchen workers would be interesting. Is there any possibility to have a different exposure level outside of hospital compared with HCWs? Higher % in assistant may help this discussion.

Reviewer 3 Report

In this excellent manuscript, the authors determine the seroprevalence of IgG and IgA antibodies in a wide array pf health care workers in a Turkish tertiary care hospital at a period of the pandemic before vaccinations were administered.  This study was well performed, carefully controlled and subjected to appropriate statistical analysis. A number of aspects of findings are extremely interesting, chief of which is that the highest percentage of seroconversion was not in HCWs with frequent and direct contact with infected patients, as perhaps one might have expected.  Rather, the highest percentage of seroconversion was surprisingly reported in kitchen workers at the hospital.  The authors attribute this to the fact that there is a constant flow of hospital employees and presumably visitors who are asymptomatic but infectious.  These individuals certainly are removing their masks while consuming food and beverages, which most assuredly accounts for the increased exposure of the cafeteria workers.  While the seroconversion levels of various HCW populations appears to be commensurate with their contact with infected patients, the kitchen worker data is most noteworthy.

Author Response

This manuscript is a resubmission of an earlier submission. The following is a list of the peer review reports and author responses from that submission.